**A discrete wavelet spectrum approach for identifying non-monotonic trend**
**patterns of hydroclimate data**
Yan-Fang Sang[1,2,3], Fubao Sun[1], Vijay P. Singh[4], Ping Xie[5], Jian Sun[1]
1. Key Laboratory of Water Cycle & Related Land Surface Processes, Institute of
Geographic Sciences and Natural Resources Research, Chinese Academy of Sciences,
Beijing 100101, China
2. Department of Atmospheric Sciences, University of Washington, Seattle 98195,
Washington, USA
3. State Key Laboratory of Hydrology-Water Resources and Hydraulic Engineering, Nanjing
Hydraulic Research Institute, Nanjing 210029, China
4. Department of Biological and Agricultural Engineering & Zachry Department of Civil
Engineering, Texas A and M University, 321 Scoates Hall, 2117 TAMU, College Station,
Texas 77843-2117, U.S.A.
5. State Key Laboratory of Water Resources and Hydropower Engineering Science, Wuhan
University, Wuhan 430072, China
**Corresponding author:**
Yan-Fang Sang: Tel/Fax: +86 10 6488 9310; E-Mail: sangyf@igsnrr.ac.cn,
sunsangyf@gmail.com
Fubao Sun: E-Mail: sunfb@igsnrr.ac.cn
**Submit to:** Hydrology and Earth System Sciences

**Abstract:** The hydroclimatic process is changing non-monotonically and identifying its trend pattern is a great challenge. Building on the discrete wavelet transform theory, we developed a discrete wavelet spectrum (DWS) approach for identifying non-monotonic trend patterns in hydroclimate time series and evaluating their statistical significance. After validating the DWS approach using two typical synthetic time series, we examined annual temperature and potential evaporation over China from 1961-2013, and found that the DWS approach detected both the "warming" and the "warming hiatus" in temperature, and the reversed changes in potential evaporation. Further, the identified non-monotonic trend patterns showed stable significance when the time series was longer than 30 years or so (i.e., the widely defined "climate" timescale). The significance of trends in potential evaporation measured at 150 stations in China, with an obvious non-monotonic pattern, was underestimated and was not detected by the Mann-Kendall test. Comparatively, the DWS approach overcame the problem and detected those significant non-monotonic trends at 380 stations, which helped understand and interpret the spatiotemporal variability of the hydroclimatic process. Our results suggest that non-monotonic trend patterns of hydroclimate time series and their significance should be carefully identified, and the DWS approach proposed has the potential for wide use in hydrological and climate sciences.

**Key words:** trend identification; discrete wavelet spectrum; decadal variability; statistical significance; Mann-Kendall test

## 1. Introduction

Climate and hydrological processes are exhibiting great variability (Allen and Ingram, 2002; Trenberth et al., 2014). Quantitatively identifying changing signals in the hydroclimate process is of great socioeconomic significance (Diffenbaugh et al., 2008; IPCC, 2013) as an important basis for hydrological modelling, understanding the future hydroclimatic regimes, and water resources planning and management. However, it remains a challenge to both scientific and social communities. The simplest and the most straightforward way to identify changes in the hydroclimate process would be to fit a monotonic (e.g., linear) trend at a certain time period at which a significance level would be assigned by a statistical test. Among the methods used for the detection of trends, the Mann-Kendall non-parametric test is most widely used and has been successfully applied in studies on climate change and its impact, when the time series is almost monotonic as required and a statistical threshold of ±1.96 is set to judge the significance of trends at 95% confidence level (Burn and Hag Elnur, 2002; Yue et al., 2002). However, due to its nonlinear and nonstationary nature, the hydroclimate process is changing and developing in a more complicated way rather than a monotonic trend way at large time scales (Cohn and McMahon, 2005; Milly et al., 2008). For example, a debate on the recent change of global air temperature has been receiving enormous public and scientific attention that the global air temperature increased during 1980-1998 passing most statistical significance tests and has since stabilized till now, widely called "global warming hiatus" (Kosaka and Xie, 2013; Roberts et al., 2015; Medhaug et al., 2017). Another known example is "evaporation paradox" (Brutsaert and Parlange, 1998; Roderick and Farquhar, 2002) that potential evaporation has worldwide declined from the 1960s, again passing most statistical significance

tests, but then reversed after the 1990s. In practice, for the hydroclimate time series, non-
monotonicity is more the rule rather than the exception (Dixon et al., 2006; Adam and
Lettenmaier, 2008; Gong et al., 2010). Therefore, identifying the non-monotonic trend pattern
hidden in those hydroclimate time series and assessing its statistical significance present a
significant research task for understanding hydroclimatic variability and changes at large time
scales.
Among those methods presently used in time series analysis, the wavelet method,
including both continuous and discrete wavelet transforms, has the superior capability of
handling nonstationary characteristics of the time series at multi-time scales (Percival and
Walden, 2000; Labat, 2005), so it may be more suitable for identifying non-monotonic trend
patterns in hydroclimate time series at large time scales. In a seminal work, Torrence and
Compo (1998) placed the continuous wavelet transform in the framework of statistical analysis
by formulating a significance test. Since then, the continuous wavelet method has become more
applicable and rapidly developed to estimate the significance of variability in climate and
hydrological studies. Especially, the continuous wavelet spectrum (i.e., continuous wavelet
variance) was established to detect those significant variabilities in the hydroclimate process
(Labat et al., 2000). However, in the continuous wavelet results of a time series, a known
technical issue is the "data redundancy" (Gaucherel, 2002; Nourani et al., 2014), which is the
redundant information across timescales leading to more uncertainty.
On the contrary, the other type of wavelet transform, i.e., the discrete wavelet transform,
has the potential to overcome that problem of data redundancy, in that those wavelets used for
discrete wavelet transform must meet the orthogonal properties. Therefore, the discrete wavelet
method can be more effective to identify and describe the non-monotonic trend pattern in a time
series (Almasri et al., 2008; de Artigas et al., 2006; Kallache et al., 2005; Partal and Kucuk,
2006; Nalley et al., 2012). However, there lacked an effective discrete wavelet spectrum in the
wavelet methodology without which uncertainty in the discrete wavelet-aided identification of
a trend could not be accurately estimated, and the significance level of the identified trend could
not be quantitatively evaluated either. For overcoming the problem, Sang et al. (2013) discussed
the definition of trend, and proposed a discrete wavelet energy function-based method for the
identification of trends, with the basic idea of comparing the difference of discrete wavelet
results between hydrological data and noise. The method used a proper confidence interval to
assess the statistical significance of the identified trend, in which the key equation for
quantifying trend's significance was based on the concept of quadratic sum. However,
computation of the quadratic sum disobeys the customary practice of computing variance in
spectral analysis. By using the quadratic sum, the significance of a non-monotonic trend cannot
be reasonably assessed, because it neglects the big influence of trend's mean value. For instance,
for those trends with small variations but big mean values, the quadratic sums are big values,
based on which the statistical significance of trends would inevitably be over-assessed.
Therefore, evaluation of the statistical significance of a non-monotonic trend in a time series
should be based on its own variability, and the influence of other factors should also be
eliminated.

By combining the advantages of discrete wavelet transform and successful practice in

spectral analysis methods, this study aimed at developing a practical but reliable discrete
wavelet spectrum approach for identifying non-monotonic trend patterns in hydroclimate time
series and quantifying their statistical significance, and further improving the understanding of
non-monotonic trends by investigating their variation with data length increase. To do that,
Section 2 presents details of the newly developed approach building on the wavelet theory and
spectrum analysis. In Section 3, we use both synthetic time series and annual time series of air
temperature and potential evaporation over China as examples to investigate the applicability
of the approach, which is followed by discussion and conclusion in the final section.
**2. A discrete wavelet spectrum approach**
Here we develop an approach, termed as "discrete wavelet spectrum approach," for
identifying non-monotonic trend patterns in hydroclimate time series, in which the discrete
wavelet transform (DWT) is used first to separate the trend pattern at large time scales, and its
statistical significance is then evaluated by using the discrete wavelet spectrum, whose
confidence interval is quantified and described through Monte-Carlo test.
Following the wavelet analysis theory (Percival and Walden, 2000), the discrete wavelet
transform of a time series $f(t)$ with a time order $t$ can be expressed as:
$$W_f(j,k) = \int_{-\infty}^{+\infty} f(t)\psi_{j,k}^{*}(t)dt \quad with \quad \psi_{j,k}(t) = a_0^{-j/2}\psi(a_0^{-j}t - b_0 k) \tag{1}$$

where $\psi^{*}(t)$ is the complex conjugate of the mother wavelet $\psi(t)$; $a_0$ and $b_0$ are constants, and
integer $k$ is a time translation factor; and $W_f(j,k)$ is the discrete wavelet coefficient under the
decomposition level $j$ (i.e., time scale $a_0^j$). In practice, the dyadic DWT is used widely by
assigning $a_0=2$ and $b_0=1$:
$$W_f(j,k) = \int_{-\infty}^{+\infty} f(t)\psi_{j,k}^{*}(t)dt \quad with \quad \psi_{j,k}(t) = 2^{-j/2}\psi(2^{-j}t - k) \tag{2}$$

The highest decomposition level $M$ is determined by the length $L$ of series $f(t)$, and can be
calculated as $log_2(L)$ (Foufoula-Georgiou and Kumar, 2014). The sub-signal $f_j(t)$ in the original
series $f(t)$ under each level $j$ ($j = 1, 2, …, M$) can be reconstructed as:

$$f_j(t) = \sum_k W_f(j,k)\psi^*(2^{-j}t - k)$$ (3)

where the sub-signal $f_j(t)$ at the highest decomposition level (when $j=M$) defines and describes
the non-monotonic trend pattern of the series $f(t)$, as generally understood. However, it should
be noted that a meaningful trend closely depends on the time scale concerned. If the variability
of series $f(t)$ on a certain smaller time scale $K$ ($K<L$) is concerned, the proper decomposition
level can be determined as $log_2(K)$, then the sum of all those sub-signals at the time scale equal
to and bigger than $K$ can be the non-monotonic trend pattern identified.
Sang (2012) discussed the influence of the choice of the mother wavelet and
decomposition level, as well as noise types on the discrete wavelet decomposition of time series,
and further proposed some methods to solve for them. By conducting Monte-Carlo experiments,
he found that the seven wavelet families (126 mother wavelets) used for DWT can be divided
into three types, and recommended the first type, by which wavelet energy functions of diverse
types of noise data keep stable and thus have little influence on the wavelet decomposition of
time series. Specifically, one chooses an appropriate wavelet, according to the relationship of
statistical characteristics among the original series, de-noised series and removed noise, chooses
a proper decomposition level by analyzing the difference between energy function of the
analyzed series and that of noise, and then identifies the deterministic components (including
trend) by conducting significance testing of DWT. These methods are closely built on the
composition and variability of hydroclimate time series at different time scales. They were used
here to accurately identify and describe the non-monotonic trend pattern in a time series, and
assess its statistical significance.

Further, to establish a reliable discrete wavelet spectrum (DWS) of time series, we need

to specify a spectrum value $E(j)$ for each sub-signal $f_j(t)$ (in Eq. 3), based on which we can
quantitatively evaluate its importance and statistical significance. Following the general
practice in conventional spectral analysis methods (Fourier transform, maximum entropy
spectral analysis, *etc.*), here we define $E(j)$ at the $j$th level by taking the variance of $f_j(t)$:

$$E(j) = \text{var}(f_j(t)) \tag{4}$$

It can accurately quantify the intensity of variation of sub-signals (including trend) by

eliminating the influence of their mean values, which is different from the quadratic sum-based
method proposed by Sang et al. (2013). For hydroclimate time series, both stochastic and
deterministic components generally have distinctive characteristics from purely noise
components (Sang et al., 2012; Rajaram et al., 2015). Due to the grid of dyadic DWT (Partal
and Cigizoglu, 2008), discrete wavelet spectra $E_r(j)$ of various noise types strictly follow an
exponentially decreasing rule with a base 2 (Sang 2012):

$$E_r(j) = 2^{-j} \tag{5}$$

The discrete wavelet spectra of deterministic components and that of noise are obviously

different. Hence, we define the DWS of noise data as the "reference discrete wavelet spectrum
(RDWS)", based on which we evaluate the statistical significance of the non-monotonic trend
pattern of a time series.
To be specific, we design a technical flowchart to show how we develop the DWS
approach for identifying the non-monotonic trend pattern of time series, and also for evaluating
the statistical significance of that trend pattern (see details in Figure 1):
(1) For the series $f(t)$ with length $L$ to be analyzed, we normalize it, and decompose it using

the DWT method in Eq. (2) and (3);

(2) We calculate the discrete wavelet spectrum of the series $f(t)$ by Eq. (4);
(3) For comparison, we then use the Monte-Carlo method to generate normalized noise data

$N$ with the same length as the series $f(t)$, and compute its RDWS by Eq. (4). Considering

that discrete wavelet spectra of diverse types of noise data consistently follow Eq. (5),

here we generate noise data following the standard normal probability distribution;

(4) We repeat the above step 5000 times, and calculate the mean value and variance of the

spectrum values (in Eq. 4) of the normalized noise data $N$ at each decomposition level $j$.

Based on it, we estimate an appropriate confidence interval of RDWS at the concerned

confidence level. In this study, we considered 95% confidence level;

(5) In comparing DWS of the series $f(t)$ and the confidence interval generated by that of

noise (i.e., RDWS), we identified the deterministic components under the highest

decomposition level as the non-monotonic trend pattern of the series, and judged

whether it was significant. Specifically, if the spectrum value of the analyzed series' sub-

signal under the highest level was above the confidence interval of RDWS, it was

considered that the non-monotonic trend pattern was statistically significant; otherwise,

if the spectrum value of the sub-signal under the highest level fell into the confidence

interval of RDWS, it was not statistically significant;

(6) If a smaller time scale $K$ is concerned, we can use the decomposition level $log_2(K)$,

instead of $M$, and then repeat the steps (1-5) to identify the non-monotonic trend pattern

at that time scale.

In the following section, we mainly investigate the applicability and reliability of the DWS

approach for identifying the non-monotonic trend and assessing its significance, and further
investigate the variation of non-monotonic trend with data length increase to improve our
understanding of trend at large time scales.
**< Figure 1>**
**3. Results**
**3.1 Synthetic series analysis**

To test and verify the reliability of the developed discrete wavelet spectrum (DWS)

approach for identifying the non-monotonic trend pattern of a time series, we considered the
general hydrological situations and generated two synthetic series data, with known signals and
noise a priori. For investigating the variation of non-monotonic trend with data length increase,
we set the length of the two synthetic series as 200, and the noise in them followed a standard
normal probability distribution. The first synthetic series S1 consisted of an exponentially
increasing line and a periodic curve (with a periodicity of 200) with some noise content (Figure
2, left panel); and the second synthetic series S2 was generated by including a hemi-sine curve,
a periodic curve (with a periodicity of 50) and some noise content (Figure 2, right panel). Using
the MK test and considering monotonic trends, series S1 showed a significant increase but the
trend of series S2 was not significant.
When using the DWS approach (Figure 1), we considered the time scale as data length,
and used the Daubechies (db8) wavelet to decompose series S1 into seven (i.e., $<log_2 200$) sub-
signals using Eq. (2) and Eq. (3). Then, we took the sub-signals under the seventh level as the
defined non-monotonic trend pattern. As shown in Figure 2 (left panel), the identified non-
monotonic trend pattern in series S1 was similar to the true trend pattern. However, the linear
fitting curve (a monotonic curve) could not capture the detail of the non-monotonic trend pattern.
The same approach applied to series S2 in Figure 2 (right panel) and the conclusion did not
change. Moreover, for series S2 with large variability at large time scales, the linear fitting
curve or other monotonic curves may not be physically meaningful.
**< Figure 2>**
We computed the discrete wavelet spectra of the two synthetic series using Eq. (4), and
used the reference discrete wavelet spectrum with 95% confidence interval to evaluate the
statistical significance of their non-monotonic trend patterns. That is, if the red point at a certain
data length was above the 95% confidence bar, described by the blue line in Figure 3, it was
considered that the trend pattern was significant at 95% confidence level. Using the DWS
approach, the trend pattern of series S1, which was quasi-monotonic, was found significant
(Figure 3a) as in the MK test (Figure 3c), but the non-monotonic series S2 showed a significant
trend pattern (Figure 3b), which was greatly different from the MK test (Figure 3d).
In Figure 3, we also presented the significance of the identified trend patterns of the two
series using both our DWS approach and the MK test, and we changed the data length to
investigate the stability of the statistical significance of the non-monotonic trend pattern.
Generally, it would have more uncertainty when evaluating the statistical significance of trend
pattern with a shorter length, corresponding to a bigger 95% confidence interval. Using our
DWS approach, the 95% confidence interval (i.e., the height of blue bars in Figure 3) for
evaluating the statistical significance of trend pattern generally decreased with the increase of
data length, as expected. However, in the MK test, the significance was always determined by
the constant thresholds of +/-1.96, regardless of the data length.
In the DWS results in Figure 3, the significance levels of non-monotonic trend patterns
did not consistently decrease with data length, but showed some fluctuation, as the proportions
of different components (including trend) in the original series varied with data length.
Furthermore, one would expect that if the trend pattern of a series at a certain length was
identified statistically significant, the trend pattern would extend with the increase of data length,
thus its significance may be more stable with a larger length of data considered. Using our DWS
approach, the trend pattern of series S1 was significant when the data length was larger than 55
(Figure 3a), being similar to the result of the MK test (Figure 3c). The trend pattern of the series
S2 was statistically significant when the data length was larger than 75 (Figure 3b). However,
using the MK test, the monotonic trend of series S2 was significant only when the data length
was between 40 and 185 (Figure 3d). In summary, the significance of trend pattern identified
by our DWS approach was more stable than that detected by the MK test, demonstrating the
advantage of the DWS approach in dealing with non-monotonic variation of hydroclimate time
series.
**< Figure 3>**
**3.2 Observed data analysis**

We used the annual time series of air temperature (denoted as TEM) and potential

evaporation (denoted as PET) over China to further verify the applicability of our developed

DWS approach for identifying non-monotonic trend patterns of a time series. These time series

were obtained from the hydroclimate data measured at 520 meteorological stations over China,

with the same measurement years from 1961 to 2013. The data have been quality-checked to

ensure their reliability for scientific research. The PET data were calculated from the Penman-

Monteith approach (Chen et al., 2005).

The average time series of TEM and PET measured at 520 stations were first considered.

Given the general nonstationary nature of observed hydroclimate time series, linear trends or

more generally monotonic curves could not capture the trend pattern with large interdecadal

variation and therefore were not particularly physically meaningful. In Figure 4 (left panel), we

presented the average annual TEM time series visually showing nonstationary characteristics

and non-monotonic variation. The TEM series decreased till the 1980s with fluctuations and

then sharply rose till the 2000s, followed by a decreasing tendency. The large fluctuation of the

average air temperature after the late 1990s is the well-known phenomenon of the "global

warming hiatus" (Roberts et al., 2015). The linear fitting curve obviously missed out the more

complicated trend pattern of the observed temperature time series. Using our DWS approach,

we decomposed the TEM series into five (i.e., $<log_2 53$) sub-signals using Eq. (2) and Eq. (3),

and took the sub-signals under the fifth level as the trend pattern, which realistically presented

the nonstationary variability of temperature at large time scales (Figure 4, left panel).

We also applied the DWS approach to the average annual PET time series. In the time

series of PET (Figure 4, right panel), there was a decreasing trend for the period from 1961 to

the 1990s, which is the well-known "evaporation paradox" leading to controversial
interpretations continuing over the last decade of hydrological cycles (Brutsaert and Parlange,
1998; Roderick and Farquhar, 2002). That decreasing trend was then followed by an abrupt
increase in around the 1990s, almost the same time when solar radiation was observed to be
reversing its trend, widely termed as "global dimming to brightening" (Wild, 2009).
Surprisingly, after the mid-2000s, PET started to decrease again (Figure 4, right panel).
Sometimes, one would propose to fit linear curves for separate time periods. Again, linear
curves could not capture the overall non-monotonic trend pattern of the PET series. Using the
same DWS approach, we identified the non-monotonic trend pattern of the PET time series
(Figure 4, right panel), which captured the two turning points of the changing trends in the
1990s and the 2000s.
**< Figure 4>**
The changes of trends in terms of magnitudes and signs for different periods led to the
difficulty in assessing and interpreting the significance of trends. For example, the PET time
series showed a significant decrease using the MK test (-3.76 < -1.96) during 1961-1992 (Figure
5d). At that moment before the reversed trend reported, the significant decrease could be
literally interpreted, as that PET had significantly declined and might be declining in the future.
However, the PET time series reversed after the 1990s and again in the 2000s, coming with an
insignificant overall trend for the whole period of 1961-2013. For the more or less monotonic
time series of the TEM series (1961-2013), the MK test detected a significant increase (6.00 >
1.96) (Figure 5c), which led to the surprise when air temperature was reported to have stopped
increasing after the late 1990s. In summary, it becomes vital to develop an approach for testing
the significance of trend pattern, which is suitable for non-monotonic time series, as it is an
important basis and a prerequisite for hydrological simulation and prediction at decadal scales.

In this study, building on the discrete wavelet transform, we proposed an operational

approach, i.e., DWS, for evaluating the significance of non-monotonic trend pattern in TEM
(Figure 5a) and PET (Figure 5b) series. For comparison purposes, we also conducted the
significance test for the two time series using the MK test (Figure 5c and 5d). Similar to Figure
3, we changed the data length to investigate the stability of statistical significance (Figure 5).
Again, results indicated that the 95% confidence interval for evaluating the statistical
significance of non-monotonic trend pattern generally decreased with data length, which was
different from the constant thresholds +/-1.96 adopted in the MK test. The significance test
using our DWS approach appeared to be more stable with data length than the MK test (Figure
5). Using our DWS approach, the trend pattern in the TEM series became significant when the
data length increased to 30, and the significance was more stable when it was greater than 35
(Figure 5a). For the case of the PET series, the trend pattern became statistically significant
when the data length was larger than 25 (Figure 5b). The findings here have important
implications for non-monotonic hydroclimate time series analysis, in that the timescale of
defining *climate* and *climate change* by the World Meteorological Organization is usually 30
years (Arguez and Vose, 2011) and in hydrological practice it is between 25-30 years.

For the whole time series investigated here, whose length was larger than 30 years, we

were able to examine the significance using the developed DWS approach. Combining the trend
pattern in Figure 4 (left panel) and the significance test in Figure 5a, we confirmed that the
trend pattern of the TEM time series from 1961-2013 identified in this study was significant at
95% confidence interval. Similarly, the trend pattern in PET was also significant (Figure 4 right
panel and Figure 5b). The significance test results suggested that the three main stages of the
series (red lines, Figure 4) were detectable as the overall trend pattern from the variability of
the series and were vital to understanding how the temperature and the PET series were
changing at interdecadal scales. In particular, the reversed change in PET and its significance
can be revealed by our DWS approach, which can provide more useful and physically
meaningful information.
**< Figure 5>**

We further detected and evaluated the significance of non-monotonic trends of the PET

time series measured at 520 stations for investigating their spatial difference. Because the trends
in the annual TEM time series were quasi-monotonic, and they were statistically significant at
most of the stations, no matter using our DWS approach or the MK test, more details of TEM
data were not repeated here. As for the trend patterns in the PET data, the results gotten from
our DWS approach (Figure 6, left panel) and those in the MK test presented substantial
differences. When conducting the statistical significance test using the MK test, the monotonic
trends were detected as significant in those the annual PET time series measured at 230 stations.
Significant downward monotonic trends were mainly found in the southern part of the Songliao
River basin, the Haihe River basin, the Huaihe River basin, some regions in South China, and
Northwest China. Significant upward monotonic trends were mainly found in the northern part
of the Songliao River basin, the upper reach of the Yellow River basin, the southwest corner of
China, and some regions in the Yangtze River Delta.
Comparatively, significant non-monotonic trends in the PET time series were detected at
380 stations throughout China. That means that those annual PET time series measured at 150
stations (28.8% of the total stations and mainly in the south part of China) mainly indicated
non-monotonic variations rather than monotonic trends at interdecadal scales, with similar
phenomena as shown in Figure 4 (right panel), and their significance was underestimated by
the MK test, which can only handle monotonic trends. Previous studies (Zhang et al., 2016;
Jiang et al., 2007) indicated that potential evaporation was influenced by more physical factors
(precipitation, air temperature, wind speed, relative humidity, etc.) in the southern part of China
rather than the northern part; thus, the potential evaporation process in South China presented
a more complex variability and was more difficult to detect and attribute its physical causes. As
a result, it is known here that the annual potential evaporation process in most parts of China
indicated significance variability at interdecadal scales, but it was underestimated by the
conventional MK test; moreover, only considering monotonic trends would cause a great
difficulty in accurately understanding the temporal and spatial variability of potential
evaporation and hydroclimate process in China, and also would be unfavorable for hydrological
predictions at interdecadal scales. Our results suggest that the non-monotonic trend pattern of
hydroclimate time series and its significance should be carefully identified and evaluated.
**< Figure 6>**
**4. Summary and Conclusion**
Climate and hydrological processes are changing non-monotonically. Identification of
linear (or monotonic) trends in hydroclimate time series, as a common practice, cannot capture
the detail of the non-monotonic trend pattern in the time series at large time scales, and then
can lead to misinterpreting climatic and hydrological changes. Therefore, revealing the trend
pattern of the time series and assessing its significance from the usually varying hydroclimate
process remains a challenge. To that end, we develop the discrete wavelet spectrum (DWS)
approach for identifying the non-monotonic trend in hydroclimate time series, in which the
discrete wavelet transform is used first to separate the trend pattern, and its statistical
significance is then evaluated by using the discrete wavelet spectrum (Figure 1). Using two
typical synthetic time series, we examine the developed DWS approach, and find that it can
precisely identify non-monotonic trend pattern in the synthetic time series (Figure 2) and has
an advantage in significance testing (Figure 3).

377        Using our DWS approach, we identify the trend pattern in the annual time series of average

temperature and potential evaporation over China from 1961-2013 (Figure 4). The identified
non-monotonic trend patterns precisely describe how temperature and PET are changing at
interdecadal scales. Of particularly interest here is that the DWS approach can help detect both
the "warming" and the "warming hiatus" in the temperature time series, and reveal the reversed
changes and the latest decrease in the PET time series. The DWS approach can provide other
aspects of information on the trend pattern in the time series, i.e., the significance test. Results
show that the trend pattern becomes more significant and the significance test becomes more
stable when the time series is longer than a certain period like 30 years or so, the widely defined
"climate" time scale (Figure 5). Using the DWS approach, in both time series of mean air
temperature and potential evaporation, the identified trend patterns are found significant (Figure
5). Moreover, significance of trend patterns in the PET time series obtained from the DWS
approach and the MK test has obviously different spatial distributions (Figure 6). The variability
of hydroclimate process at large time scales, especially for non-monotonic trend patterns, would
be underestimated by the MK test, which causes a great difficulty in understanding and
interpreting the spatiotemporal variability of hydroclimate process. Comparatively, the
developed DWS approach can quantitatively assess the statistical significance of non-
monotonic trend pattern in the hydroclimate process, and so can meet practical needs much
better.
In summary, our results suggest that the non-monotonic trend pattern of hydroclimate time
series and its statistical significance should be carefully identified and evaluated, and the DWS
approach developed in this study has the potential for wider use in hydrological and climate
sciences.
**Acknowledgments**
The authors gratefully acknowledged the valuable comments and suggestions given by the
Editor and the anonymous reviewers. The observed data used in the study was obtained from
the China Meteorological Data Sharing Service System (http://cdc.cma.gov.cn/). This study
was financially supported by the National Natural Science Foundation of China (No. 91647110,
91547205, 51579181), the Program for the "Bingwei" Excellent Talents from the Institute of
Geographic Sciences and Natural Resources Research, CAS, the Youth Innovation Promotion
Association CAS (No. 2017074), and the Open Foundation of State Key Laboratory of
Hydrology-Water Resources and Hydraulic Engineering (No. 2015491811).

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

**Figures**



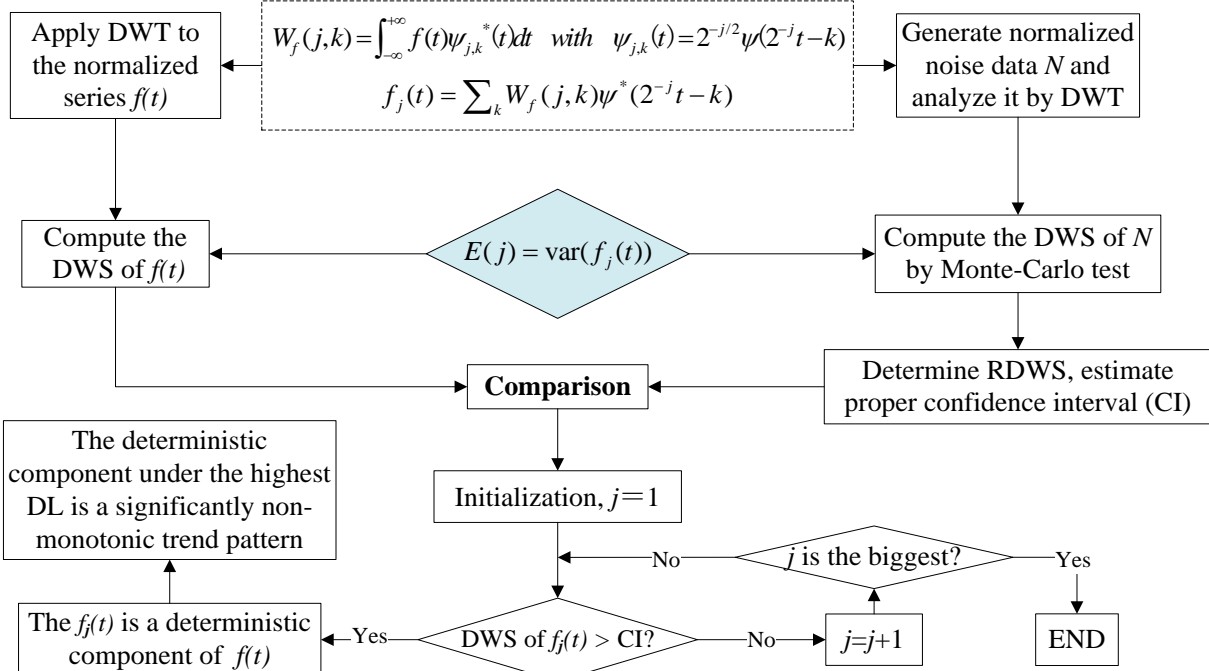


**Figure 1.** Technical flowchart for identification of the non-monotonic trend pattern in a time
series using the discrete wavelet spectrum (DWS) approach developed, where the discrete
wavelet transform (DWT) method was used to decompose the time series, and the reference
discrete wavelet spectrum (RDWS) with certain confidence interval (CI) was used for the
evaluation of significance.


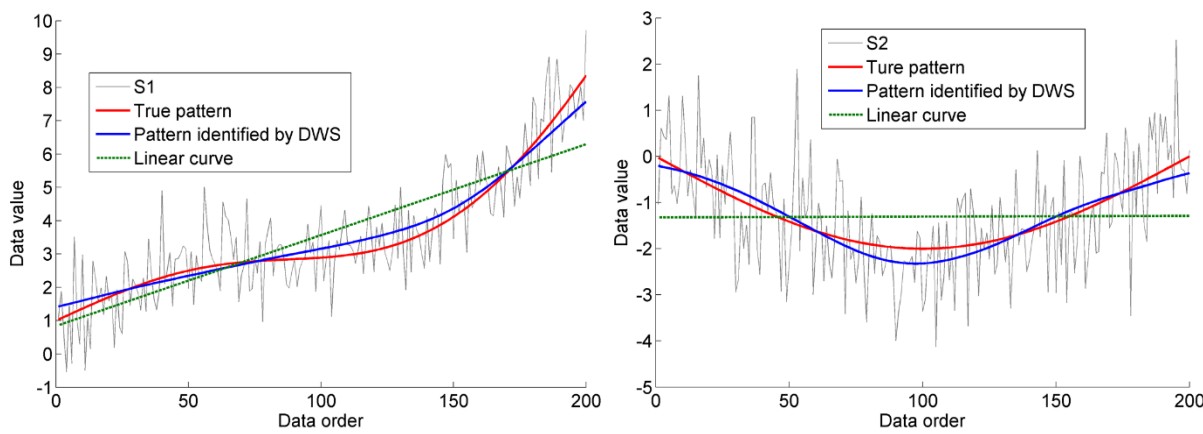


**Figure 2.** Non-monotonic trend patterns in the synthetic series S1 and S2 identified by the
discrete wavelet spectrum (DWS) approach, and the linear trends in the two series.
Synthetic series S1 is generated as: $S1=1.112^{0.1t}+0.8\times\sin(0.01\pi t)+\alpha$; and synthetic series S2
is generated as: $S2=\sin(0.04\pi t)+2\times\sin(\pi+0.005\pi t)+\alpha$, where $\alpha$ is a random process
following the standard normal distribution.




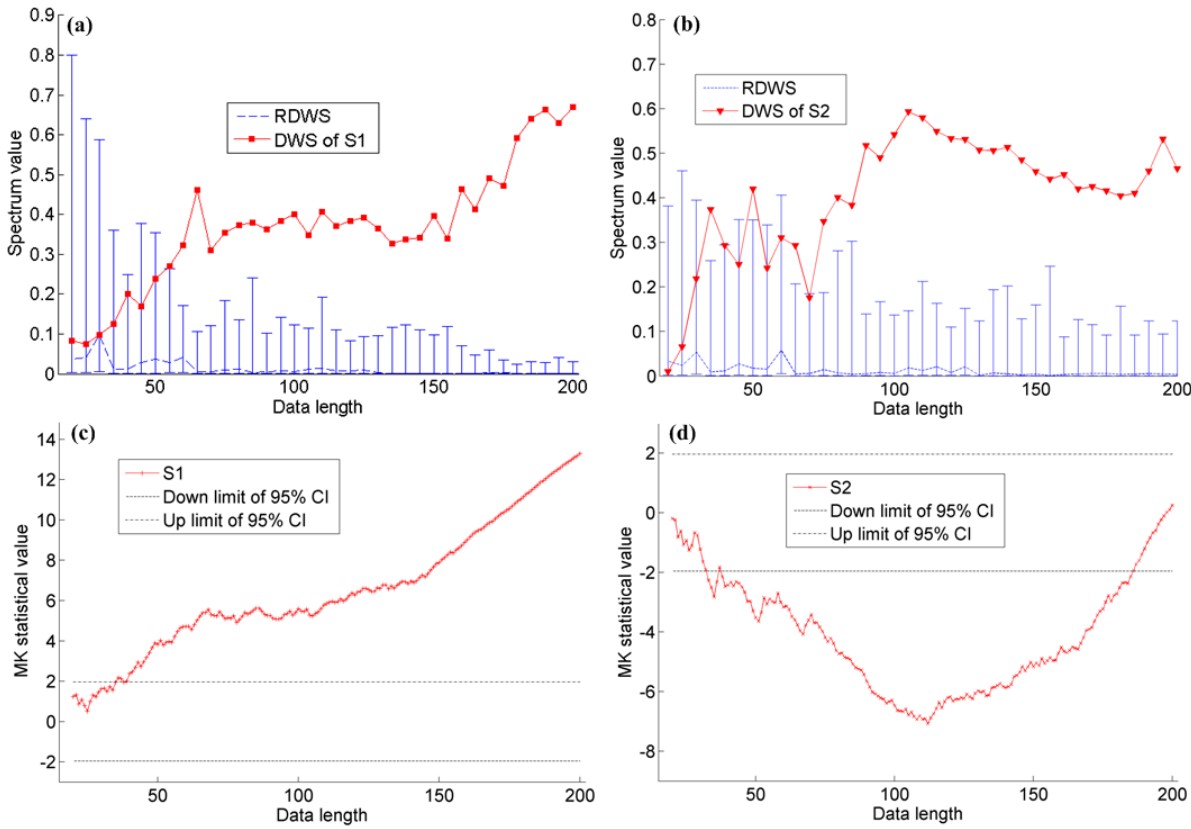

**Figure 3.** Evaluation of statistical significance of non-monotonic trend patterns in the synthetic

series S1 (a) and S2 (b) with different data length by the discrete wavelet spectrum (DWS)

approach, and the results by the Mann-Kendall (MK) test (c and d). In figure a and b, the

blue line is the reference discrete wavelet spectrum (RDWS) with 95% confidence interval

under each data length; if the red point at certain data length is above the blue bar, it is

thought that the trend pattern is significant at 95% confidence level. In figure c and d, the

two black dash lines indicate 95% confidence interval (CI) with the thresholds of +/- 1.96

in the MK test.

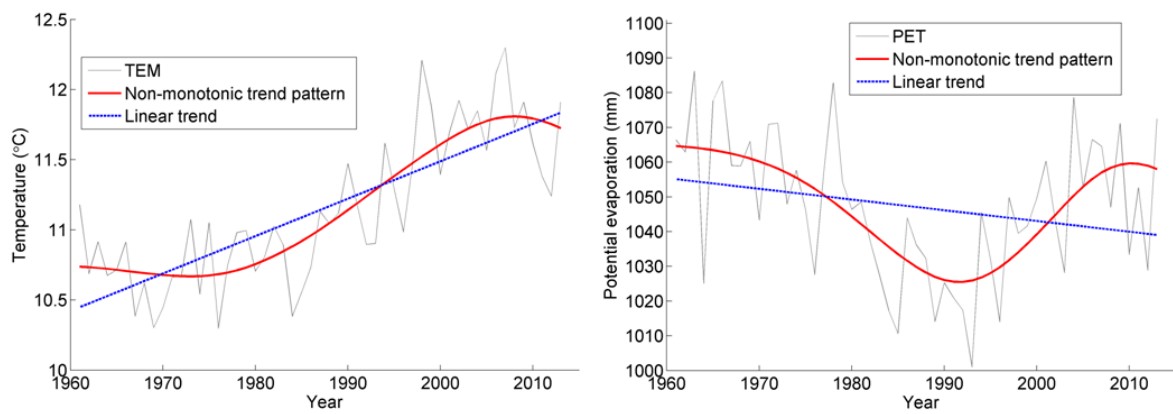


**Figure 4.** Non-monotonic trend patterns in the annual time series of the mean air temperature
(TEM) and the potential evaporation (PET) over China from 1961-2013 identified by the
discrete wavelet spectrum (DWS) approach, and the linear trends in the two series.





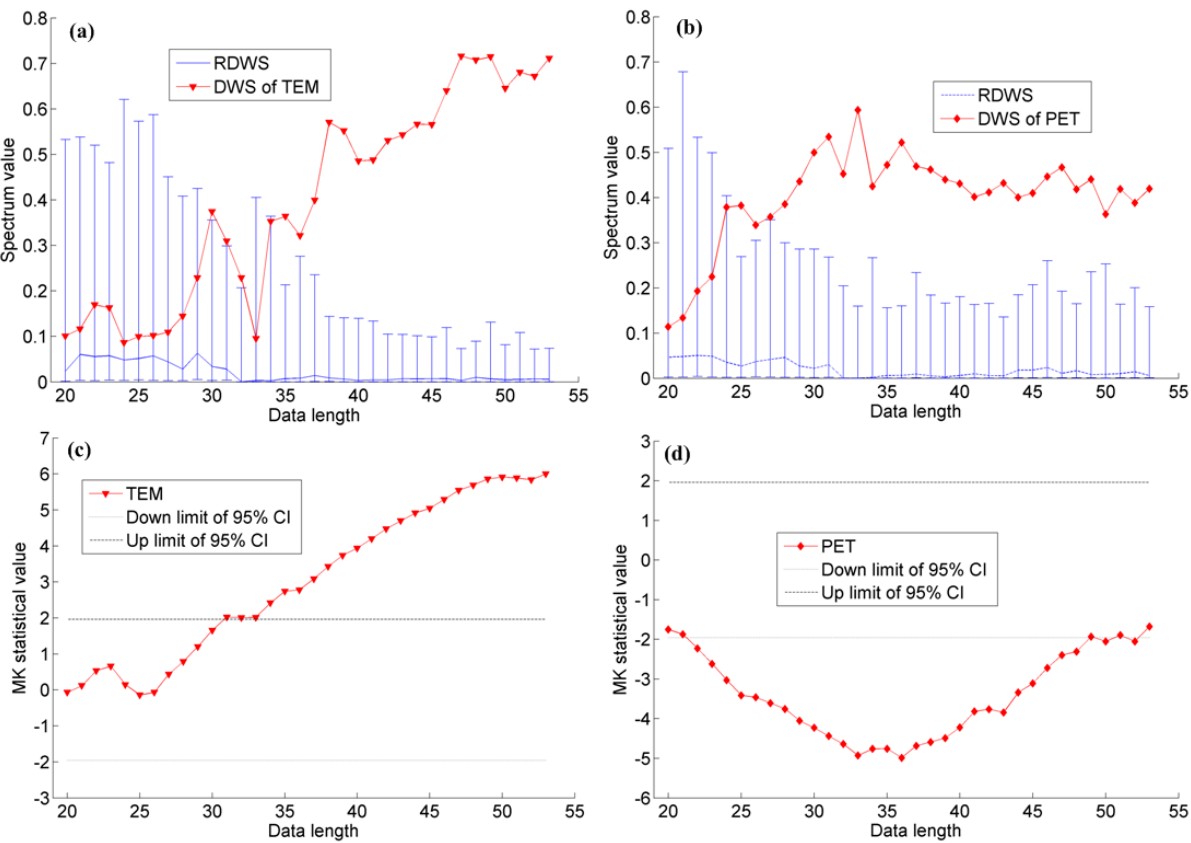


**Figure 5.** Evaluation of statistical significance of non-monotonic trend patterns in the annual

time series of the mean air temperature (TEM, a) and the potential evaporation (PET, b)

over China with different data length by the discrete wavelet spectrum (DWS) approach,

and the results by the Mann-Kendall (MK) test (c and d). In figure a and b, The blue line is

the reference discrete wavelet spectrum (RDWS) with 95% confidence interval under each

data length; and in figure c and d, the two black dash lines indicate 95% confidence interval

(CI) with the thresholds of +/- 1.96 in the MK test.






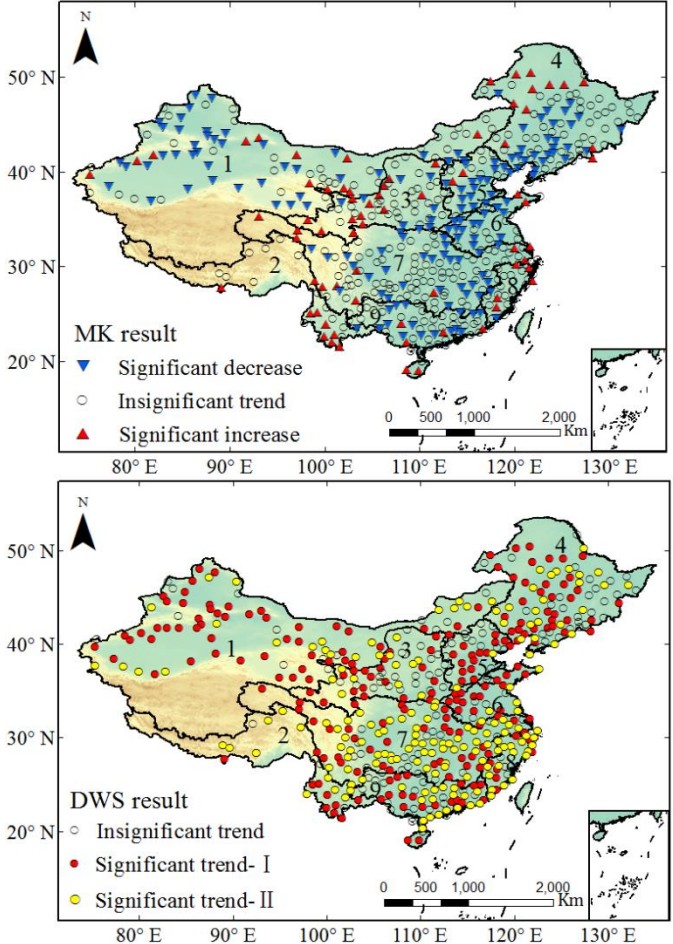


**Figure 6.** Spatial distribution of the significance of trends in the annual potential evaporation
data during 1961-2013 and measured at 520 weather stations over China. The result above
was gotten from the Mann-Kendall (MK) test. The result below was gotten from the
discrete wavelet spectrum (DWS) approach developed, in which significant trend-I means
those significant trends (at 230 stations) can be identified by both the DWS approach and
the MK test, but significant trend-II means those significant trends (at 150 stations) can
only be identified by the DWS approach but not the MK test. 1, the Northwest Inland River
basin; 2, the Southwest River basin; 3, the Yellow River basin; 4, the Songliao River basin;
5, the Haihe River basin; 6, the Huaihe River basin; 7, the Yangtze River basin; 8, the
Southeast River basin; and 9, the Pearl River basin.