# Peer review of "1. Introduction"

_Hydrology and Earth System Sciences, 2017_

## Short Comment (SC1) · 10 Feb 2017

The authors stated that they developed and applied a method called DWS to identify non-monotonic trend and its significance. As it is very well known in time series where there exists a non-monotonic trend, times between failures alternate between increasing and decreasing trend (cyclic) or have a decreasing trend, no trend, and then increasing trend. Taking in to consideration this nature of a non-monotonic trend, I personally believe that a simple visual inspection could help identify the presence of non-monotonic trend. I do not think following the process depicted in Fig. 1 is necessary at all. As to testing the level of significance, the well-known Anderson-Darling test rejects the null hypothesis in the presence of both monotonic and non-monotonic

trends.

---

## Short Comment (SC2) · 12 Feb 2017

Y.-F. Sang

sangyf@igsnrr.ac.cn

Thanks very much for giving this helpful comment. As is well known, hydrological and climate process usually present non-monotonic trend at long-term time scales. Thereby, how to accurately describe the trend and assess its statistical significance are two important issues for the detection of trends in hydroclimate data. Of course we can visually check the non-monotonic trend in a time series; by determining the break-points, we can use certain functions to describe the trend between two breakpoints. However, the approach cannot meet the practical needs enough, by which the detected trend is just composed by some segments. More importantly, we need to continuously describe the change of non-monotonic trend with time, because it is an important basis

for the simulation and prediction of decadal variability of hydroclimate process. Furthermore, the significance of a trend depends on both its own magnitude and its ratio in the original time series, and thus assessing the significance of a trend should consider the variability of the trend and the influence of other components in original series. In this article, we proposed the DWS method to identify the non-monotonic trend in hydroclimate data. We used the discrete wavelet method mainly to separate and describe the trend component, which shows continuous and smooth variation; then, we used the established discrete wavelet spectrum to assess its significance, which directly reflect its variability degree and its ratio in original time series. Therefore, we think that the proposed DWS is reliable and can be used in climate change detection and attribution studies. The Anderson-Darling test, as mentioned in the comment, is used widely for assessing whether a data set fits certain probability distribution or not. However, we don't think it can meet our need for the detection of trend, as mentioned above.

---

## Referee Comment (RC1) · Anonymous Referee #1 · 19 Apr 2017

Thank you for the opportunity to review "A discrete wavelet spectrum approach to identifying non-monotonic trend pattern of hydroclimate data" by Sang et al. The paper indicates that non-monotonic trends found by a discrete wavelet spectrum (DWS) approach can have higher significance than the trends identified by a Mann-Kendall (MK) test. The paper also demonstrates the DWS approach using two synthetic time series that have a lower-frequency periodic component and higher-frequency "noise", and time series of temperature and potential evaporation in China.

The method section seems to be the main contribution of this paper, but it is a bit terse and would be challenging for someone not familiar to wavelets to understand the approach. Wavelets are described in many papers and textbooks, but the use of wavelets

to identify trends is not common in hydrology. It would be helpful to provide the reader with more background information so the reader can understand why certain decisions are being made here. That is, methods have few equations and have short statements of the assumptions that go into the choice of equations. The following comments identify specific locations of the text where the reader could use more information on the methods.

Line 121: It would be helpful to the reader to provide background on decomposition M and why this is important for identifying a trend. Please consider adding some background on the decomposition level and why the largest level has a temporal scale that is L, the length as the time series. More specifically, why the largest level could be considered to be a trend. It could be noted that a smaller temporal scale could be important, and the decomposition level can be calculated as log2(T) if T is a temporal scale other than L, the length of the time series.

Line 126: Can you indicate which wavelet is used in this analysis?

Line 140: The text says that a spectrum is needed. Can you explain why E(j) is needed for each sub-signal?

Here are comments on various parts of the text.

Line 82: This is a good opportunity to add references to prior studies that document the DWT approach for trend estimation.

Line 90: Can you add more description about which common practice is disobeyed?

Line 135: Other studies have described similar approaches to identify a deterministic trend using DWT (e.g. Kallache et al., 2005). Could a stochastic component be added using the framework presented here?

Line 137: This statement is subjective. Can you add references here to show why you are assuming that these methods are reliable and reasonable? What is your criteria for what is reliable and reasonable?

Line 145: Please consider omitting the word "obviously." This is subjective and the result may not be obvious to everyone.

Line 180: This statement has no supporting information. Please consider deleting this sentence.

Line 227: Can you provide an explanation of why the DWT approach has a different level of significance for different data lengths than the MK approach? The benefit of the DWT approach doesn't seem to be fully explained unless you describe the reason for it to be more stable than MK.

Various parts of the text say that a result is "interesting." Please try to omit this term, and let the reader decide which results are interesting.

Figure 1: Please consider adding the numbered steps from lines 159 to 178 to the flow chart. It may be difficult for some readers to relate the numbered steps to the steps in the flow chart. Why are the DWT equations shown at the top of the flow chart? These equations are already part of the first step on the upper left of the flow chart. Figure 3, can you provide more guidance on how to assess the significance at different data lengths? It appears that the DWS is significant when it plots above the 95% confidence bar in the blue lines. Can you provide more guidance?

References: Please add publication year to each reference.

———————————————————

---

## Referee Comment (RC2) · Anonymous Referee #2 · 12 May 2017

The authors present a discrete wavelet spectrum (DWS) approach to identify the non-monotonic trend in the temperature and potential evaporation time series and to quantify the statistical significance with significance testing. The paper is well written and the research is within the scope of HESS. However, I don't think it meets the standard to be published in HESS due to below reasons.

1. DWS is a well established approach and has been widely applied, especially in signal analysis. I simply cannot see the novelty despite the authors stated they developed a new DWS approach. The novelty should be further elaborated and highlighted should the authors consider to revise and resubmit to another journal.

2. Application of DWS is limited on time series trend identification. Data interpretation

is indeed important to understand the hydro-climate system. However, it would be much practically useful if the application can be extended to trend/data forecasting.

3. The analyzed hydro-climate data are averaged time series over 740 meteorological stations over China, if I understand correctly. By averaging, the features related to different climatic regimes. geological characteristics and geographical locations, etc. will be filtered out. To analyze the time series with different features would be of more interest and revealing than just to analyze the averaged data. Also, I don't think a time series with 53 annual value is long enough to detect the reliable trend.

————————————————

---

## Author Comment (AC1) · 17 May 2017

To: Editor, Hydrology and Earth System Sciences Subject: Revised manuscript (#hess-2017-6) The Authors: Sang Y.F., et al. Title: A discrete wavelet spectrum approach to identifying non-monotonic trend pattern of hydroclimate data

Response:

The authors appreciate the Editor and Reviewers for helpful and constructive comments that improved our original manuscript submitted to HESS.

Response to Reviwer#1's comments: Comment 1: The method section seems to be the main contribution of this paper, but it is a bit terse and would be challenging for

someone not familiar to wavelets to understand the approach. Wavelets are described in many papers and textbooks, but the use of wavelets to identify trends is not common in hydrology. It would be helpful to provide the reader with more background information so the reader can understand why certain decisions are being made here. That is, methods have few equations and have short statements of the assumptions that go into the choice of equations. The following comments identify specific locations of the text where the reader could use more information on the methods.

Reply: Thanks very much for giving these valuable comments and suggestions, which is very helpful for improving the study results of our paper. The DWS approach proposed is the main contribution in our paper. Following the comment, we have rewritten and added many contents in Section 2, especially about the determination of proper decomposition level and the main technical steps of trend identification using the DWS approach, mainly to make the proposed approach more understandable. Besides, we also added some new references in the revised manuscript. More details can be found in the following point-to-point response.

Comment 2: It would be helpful to the reader to provide background on decomposition M and why this is important for identifying a trend. Please consider adding some background on the decomposition level and why the largest level has a temporal scale that is L, the length as the time series. More specifically, why the largest level could be considered to be a trend. It could be noted that a smaller temporal scale could be important, and the decomposition level can be calculated as log2(T) if T is a temporal scale other than L, the length of the time series.

Reply: Thanks very much for giving this valuable comment. The decomposition level j just reflects the time scale (i.e., time scale a0j in Eq. 2) for wavelet analysis, which is important to identify the trend pattern in a hydroclimate time series. Following the comment, we further explain the key point about the decomposition level by adding some contents in lines 121 and 124-125 and adding a new reference.

As generally considered, the variation of a time series at the biggest time scale (i.e., its data length) reflects its trend, thus in practice, the data length is usually chosen at the time scale for the trend identification. We also know that the time scale is an important factor for trend identification, and if those variations at smaller time scales are concerned, we can change the proper decomposition level in Eq. (2) and then get the trend pattern. Following the valuable comment, we added some contents as "However, it should be noted that a meaningful trend closely depends on the temporal scale concerned. If the variability of series f(t) on certain smaller time scale K (K<L) is concerned, the proper decomposition level can be determined as log2(K); then, the sum of those sub-signals at the time scales bigger M can be the non-monotonic trend pattern identified." in lines 129-133 in the revised manuscript.

Comment 3: Line 126: Can you indicate which wavelet is used in this analysis?

Reply: Thanks. Following the helpful comment, in the revised manuscript we explained the approach for the choice of proper wavelet in lines 139-141. Further, the wavelet used for the analysis of time series in the study was added in line 208.

Comment 4: Line 140: The text says that a spectrum is needed. Can you explain why E(j) is needed for each sub-signal?

Reply: Thanks. Because we want to establish a reliable discrete wavelet spectrum for assessing the statistical significance of trend pattern in a time series, in the revised manuscript we defined the E(j) as a spectrum value at each decomposition level j. As explained in lines 155-159, the discrete wavelet spectra Er(j) of various noise types strictly follow an exponentially decreasing rule with base 2 along with the decomposition level increase, which is obviously different from that of hydroclimate time series, thus we can take the former as a basis for establishing the discrete wavelet spectrum approach.

Following the favorable comment, we added some contents as: "to establish the discrete wavelet spectrum (DWS) of time series, we need to specify a spectrum value E(j)

for each sub-signal fj(t) (in Eq. 3), based on which we can quantitatively evaluate its importance and statistical significance." in lines 147-149 in the revised manuscript.

Comment 5: Line 82: This is a good opportunity to add references to prior studies that document the DWT approach for trend estimation.

Reply: Thanks. We have added five new references in lines 84-85 in the revised manuscript.

Comment 6: Line 90: Can you add more description about which common practice is disobeyed?

Reply: Thanks. Following the helpful comment, we rewrote the sentence as "However, the practice of quadratic sum disobeys the common practice of computing variance in spectral analysis, and sometimes cannot reasonably assess the significance of non-monotonic trend" in lines 93-95 in the revised manuscript.

Comment 7: Line 135: Other studies have described similar approaches to identify a deterministic trend using DWT (e.g. Kallache et al., 2005). Could a stochastic component be added using the framework presented here?

Reply: Thanks. In the revised manuscript we mainly proposed the DWS approach for identifying the trend pattern in a hydroclimate time series. To be specific, we used the discrete wavelet decomposition method to separate the trend pattern in a time series, and more importantly, we then established the discrete wavelet spectrum to assess its statistical significance, which is the novelty of this study and is different from previous studies.

Hydroclimate time series is generally composed of deterministic components, stochastic components and noise, and identification of the significance of a trend pattern is just to judge if it is a deterministic component. As explained above, in the revised manuscript we defined the E(j) as a spectrum value at each decomposition level j, and found that the discrete wavelet spectra E(j) of various noise types strictly follow

an exponentially decreasing rule with base 2, which is obviously different from that of deterministic and stochastic components in hydroclimate time series, thus we took the former as a basis for establishing the discrete wavelet spectrum approach. That is, a stochastic component can be added using the framework presented in our study, and it would not influence the identification of trend pattern and its statistical significance.

Comment 8: Line 137: This statement is subjective. Can you add references here to show why you are assuming that these methods are reliable and reasonable? What is your criteria for what is reliable and reasonable?

Reply: Thanks. Following the favorable comment, these subjective statements were removed in the revised manuscript.

Comment 9: Line 145: Please consider omitting the word "obviously." This is subjective and the result may not be obvious to everyone.

Reply: Thanks. The subjective word "obviously" was removed in line 154 in the revised manuscript.

Comment 10: Line 180: This statement has no supporting information. Please consider deleting this sentence.

Reply: Thanks. These inaccurate statements were removed in the revised manuscript.

Comment 11: Line 227: Can you provide an explanation of why the DWT approach has a different level of significance for different data lengths than the MK approach? The benefit of the DWT approach doesn't seem to be fully explained unless you describe the reason for it to be more stable than MK.

Reply: Thanks very much for giving this valuable comment. In our opinion, the significance of a trend pattern is determined by both its own magnitude and its proportion in the original time series, and the significance of a trend pattern would change with data length, because the proportions of different components (including trend) in the original series vary with data length.

Following the comment, we added some new contents to more clearly explain why the significance level of a trend pattern varies with data length. To be specific, we explained that "Generally, it would have more uncertainty when evaluating the statistical significance of trend pattern with a shorter length, corresponding to a bigger 95% confidence interval" in lines 227-229 and "the significance levels of the trend patterns do not consistently decrease with data length, but show some fluctuation, as the proportions of different components (including trend) in the original series vary with data length" in lines 234-236.

Comment 12: Various parts of the text say that a result is "interesting." Please try to omit this term, and let the reader decide which results are interesting.

Reply: Thanks. The inaccurate word "interesting" was removed or changed as a more accurate word throughout the manuscript.

Comment 13: Figure 1: Please consider adding the numbered steps from lines 159 to 178 to the flow chart. It may be difficult for some readers to relate the numbered steps to the steps in the flow chart. Why are the DWT equations shown at the top of the flow chart? These equations are already part of the first step on the upper left of the flow chart. Figure 3, can you provide more guidance on how to assess the significance at different data lengths? It appears that the DWS is significant when it plots above the 95% confidence bar in the blue lines. Can you provide more guidance?

Reply: Thanks very much. Following the valuable comment, we divided the analysis process of trend identification using the DWS approach into five steps, and more details can be found in lines 168-190. It is just for making the Figure 1 presentable by putting the DWT equation on the top. Following this comment, the Figure 1 was carefully readjusted.

Moreover, following the valuable comment, we added some new contents to explained the results in Figure 3 as "That is, if the red point at certain data length is above the 95% confidence bar, described by the blue line in Figure 3, it is thought that the trend

pattern is significant at 95% confidence level" in lines 219-221, and in its caption.

Comment 14: References: Please add publication year to each reference.

Reply: Thanks. We have checked all the references and make sure that all references have publication year.

Thank you very much!

Best Regards! Yan-Fang Sang

Please also note the supplement to this comment:
http://www.hydrol-earth-syst-sci-discuss.net/hess-2017-6/hess-2017-6-AC1-supplement.pdf

---

## Author Comment (AC2) · 17 May 2017

To: Editor, Hydrology and Earth System Sciences Subject: Revised manuscript (#hess-2017-6) The Authors: Sang Y.F., et al. Title: A discrete wavelet spectrum approach to identifying non-monotonic trend pattern of hydroclimate data

Response:

The authors appreciate the Editor and Reviewers for helpful and constructive comments that improved our original manuscript submitted to HESS.

Response to Reviwer#2's comments: Comment 1. DWS is a well established approach and has been widely applied, especially in signal analysis. I simply cannot see the

novelty despite the authors stated they developed a new DWS approach. The novelty should be further elaborated and highlighted should the authors consider to revise and resubmit to another journal.

Reply: Thanks for giving this helpful comment. We know that the discrete wavelet transform method has been widely used for trend identification; however, how to accurately assess the statistical significance of a trend pattern gotten from discrete wavelet decomposition results is a big challenge. To solve the problem, in the revised manuscript we mainly proposed the DWS approach for identifying the trend pattern in a hydroclimate time series. To be specific, we used the discrete wavelet decomposition method to separate the trend pattern in a time series, and more importantly, we then established the discrete wavelet spectrum to assess its statistical significance, which is the novelty of this study and is different from previous studies.

Following the helpful comments given by both the Reviewer 1 and 2, we have carefully rewritten and added many contents in the revised manuscript, and added some new references about the use of DWT for trend identification, mainly to more clearly explain the DWS approach proposed and emphasize the advantage of the approach.

Comment 2. Application of DWS is limited on time series trend identification. Data interpretation is indeed important to understand the hydro-climate system. However, it would be much practically useful if the application can be extended to trend/data forecasting.

Reply: Thanks for giving this favorable comment. Identification of trend is an important issue to understand the variability of hydroclimate process at long time scale, but it is not an easy task in practice. In the revised manuscript we mainly proposed the DWS approach for identifying the trend pattern in hydroclimate time series and assessing its significance.

Of course data simulation and forecasting is an important scientific issue and task to understand the future situations, and these have been many relevant studies, but in

our opinion, accurate identification of the variability (including trend) of hydroclimate time series is the basic and primary task for the data interpretation and forecasting. Therefore, in the study we focused on the issue of trend identification, as an important basis of data forecasting at mid-to-long time scales.

Comment 3. The analyzed hydro-climate data are averaged time series over 740 meteorological stations over China, if I understand correctly. By averaging, the features related to different climatic regimes. geological characteristics and geographical locations, etc. will be filtered out. To analyze the time series with different features would be of more interest and revealing than just to analyze the averaged data. Also, I don't think a time series with 53 annual value is long enough to detect the reliable trend

Reply: Thanks very much. As explained above, in this study we mainly proposed the DWS approach for identifying the trend pattern in hydroclimate time series and assessing its statistical significance, but not investigating the variability of spatial-temporal variability of temperature and precipitation process over China. We used the TEM and PET time series mainly for verifying the applicability of proposed approach by analyzing the "warming" and the "warming hiatus" in temperature, and the reversed changes in potential evaporation, which cannot be described by a monotonic trend.

Besides, we consider that the identification of trend in a shorter time series would have more uncertainty, so in this study we proposed DWS approach just for assessing the statistical significance of trend pattern, which is described by a proper confidence interval in Figure 3 and 5. Those observed hydroclimate time series in China are usually about 50 years, by using which we can investigate their variability at decadal and multi-decadal time scales. Of course we know that a meaningful trend closely depends on the temporal scale concerned, and the proposed DWS approach can be used for identifying the trend pattern at different time scales. To clarify the point, we added some new contents in lines 129-133.

Thank you very much!

Best Regards! Yan-Fang Sang

Please also note the supplement to this comment:
http://www.hydrol-earth-syst-sci-discuss.net/hess-2017-6/hess-2017-6-AC2-supplement.pdf

---

## Referee Report (RR1)

Thank you for addressing the comments of the reviewers. It appears that the comments are sufficiently addressed, and the paper is improved.

---

## Referee Report (RR2)

Thank you for correcting the manuscript according to the comments of the reviewers. Paper is now sufficiently improved.

---

## Author Response (AR2)

**To: Editor, Hydrology and Earth System Sciences**
**Subject:** Revised manuscript (#hess-2017-6)
**The Authors:** Sang Y.F., et al.
**Title:** A discrete wavelet spectrum approach to identifying non-monotonic trend pattern of hydroclimate data

**Response:**

The authors appreciate the Editor and Reviewers for helpful and constructive comments that improved our original manuscript submitted to HESS.

**Response to Editor's comments:**

*While Reviewer #1 felt that the paper could be published "as is", Reviewer #2 did not feel that revision to address the novelty of this paper's contribution was addressed in a sufficient way. After looking over the revised track-changes manuscript, the authors' proposed responses, and the reviewer comments, I concur with Reviewer #2. For example, the authors noted in their proposed responses that they would, "carefully rewrite and add many contents in the revised manuscript, and add some new references about the use of DWT for trend identification, mainly to more clearly explain the DWS approach proposed and emphasize the advantage of the approach." In examining the track changes version of the revised discussion paper, the only change I was able to observe is that a few more references have been added to the introduction. Since neither reviewer recommended rejection, I will not overrule this recommendation. However, I do hope to see the comments of Reviewer #2 fully addressed in this next revision. The discussion paper will once again be sent back out for review before making a final decision on publication.*

**Reply:** Thank you very much for giving these helpful comments. We have revised the manuscript following your comments and suggestions. As you can find in the revised manuscript, we added some new contents to clarify the novelty of the study. To be specific, we added those contents in lines 81-83 to describe the wide use of the continuous wavelet transform and the continuous wavelet spectrum in hydrology studies, and further added those contents in lines 91-94 to emphasize the lack of an effective discrete wavelet spectrum approach in the wavelet methodology: "*However, there lacked an effective discrete wavelet spectrum in the wavelet methodology. Without it, uncertainty in the discrete wavelet-aided identification of trend cannot be accurately estimated, and the significance level of the identified trend cannot be quantitatively evaluated, either*". Therefore, we proposed the discrete wavelet spectrum (DWS) approach for detecting non-monotonic trends in hydroclimate time series, as an important basis of understanding the variability of hydroclimate process at large time scales.

Moreover, we added those contents in lines 339-370 and the new Figure 6 to describe the spatial distribution of significance of trends in potential evaporation over China, and found the big different results gotten from the DWS approach proposed and the Mann-Kendall (MK)

test. We found that the MK test underestimated the significant of those trends (at 150 stations among total 520 stations) with non-monotonic variations, which is unfavorable for accurately understanding the temporal and spatial variability of potential evaporation and hydroclimate process in China. Therefore, we think that the DWS approach performs better for detection of non-monotonic trends in hydroclimate time series, and it is the novelty of this study.

More details can be found in the following point-to-point response.

**Response to Reviwer#2's comments:**

*Comment 1. DWS is a well established approach and has been widely applied, especially in signal analysis. I simply cannot see the novelty despite the authors stated they developed a new DWS approach. The novelty should be further elaborated and highlighted should the authors consider to revise and resubmit to another journal.*

**Reply:** Thanks for giving this helpful comment. We know that the wavelet methodology, including both continuous and discrete wavelet transform, has been widely used for hydrology studies. And the continuous wavelet spectrum (i.e., continuous wavelet variance) was also established to detect those significant variabilities in the hydroclimate process. However, there is a "data redundancy" problem in continuous wavelet transform. Comparatively, the discrete wavelet transform can overcome the problem, and can also describe the trend pattern using those sub-signals of the original series at large time scales, so it can be more suitable for trend identification. However, there lacked an effective discrete wavelet spectrum in the wavelet methodology. Without it, uncertainty in the discrete wavelet-aided identification of trend cannot be accurately estimated, and the significance level of the identified trend cannot be quantitatively evaluated, either. Therefore, we proposed the DWS approach in the manuscript, and it is the main novelty of the study.

Following the helpful comments, we added those contents in lines 81-83 to describe the wide use of the continuous wavelet transform and the continuous wavelet spectrum in hydrology studies, and then added those contents in lines 88-94 to emphasize the lack of an effective discrete wavelet spectrum approach in the wavelet methodology. It is the incentive of proposing the DWS approach in the study. Moreover, we added those contents in lines 339-370 (new Figure 6) to more clearly verify the better performance of the DWS approach compared with the MK test used widely for trend identification, and also to describe the necessity of the study for detecting non-monotonic trends.

*Comment 2. Application of DWS is limited on time series trend identification. Data interpretation is indeed important to understand the hydro-climate system. However, it would be much practically useful if the application can be extended to trend/data forecasting.*

**Reply:** Thanks for giving this favorable comment. We agree the opinion, that is, detection of variability (including trend) in hydroclimate process is an important basis of hydrological simulation and prediction at large time scales, as a practical guide for water resources planning and management. Considering that hydrological prediction is another big issue and is related to many other issues, we didn't discuss it too much here. However, following the favorable comment, we added some contents in the manuscript (in lines 49-50, 308-309 and 367-368) to briefly clarify the importance of the issue and its relationship with this study.

*Comment 3. The analyzed hydro-climate data are averaged time series over 740 meteorological stations over China, if I understand correctly. By averaging, the features related to different climatic regimes. geological characteristics and geographical locations, etc. will be filtered out. To analyze the time series with different features would be of more interest and revealing than just to analyze the averaged data. Also, I don't think a time series with 53 annual value is long enough to detect the reliable trend*

**Reply:** Thanks very much for giving the valuable comment. Following the comment, we added those contents in lines 339-370 and the new Figure 6 to describe the spatial distribution of significance of trends in potential evaporation (PET) in China, based on which we further verify the better performance of the DWS approach proposed.

Interestingly, we found that when using the MK test, the monotonic trends were detected as significant in those annual PET time series measured at 230 stations (in lines 344-351), however, the significant non-monotonic trends in PET time series can be detected at 380 stations throughout China. That means, those annual PET time series measured at 150 stations (28.8% of the total stations and mainly in the south part of China) mainly indicated non-monotonic variations rather than monotonic trends at interdecadal scales, with similar phenomena as shown in Figure 4 (right panel), and their significance was underestimated by the MK test (in lines 352-357). Following previous studies, we know that potential evaporation process was influenced by more physical factors (precipitation, air temperature, wind speed, relative humidity, etc) in the south part of China rather than the north part; thus, potential evaporation process in South China presented more complex variability, and was more difficult to detect and attribute its physical causes (in lines 357-362).

Therefore, from the results in Figure 6 we can further verify the better performance and effectiveness of the DWS approach proposed for the detection of non-monotonic trends in hydroclimate time series, and suggest that the non-monotonic trend pattern of hydroclimate time series and its significance should be carefully identified and evaluated.

Besides, we think that detection of trend is closely related to the time scales concerned. In this study, we mainly used the observed hydroclimate data with 53 years to investigate the variability of TMP and PET process at interdecadal scales.

Thank you very much!

Best Regards!
Yan-Fang Sang

[revised manuscript text omitted]

---

## Author Response (AR3)

**To: Editor, Hydrology and Earth System Sciences**
**Subject:** Revised manuscript (#hess-2017-6)
**The Authors:** Sang Y.F., et al.
**Title:** A discrete wavelet spectrum approach to identifying non-monotonic trend pattern of hydroclimate data

**Response:**

The authors appreciate the Editor for helpful and constructive comments that improved our original manuscript submitted to HESS.

**Response to Editor's comments:**

*The manuscript has not received 3 additional reviews. All reviews recommend that the paper is a useful contribution and that previous reviewer comments have been sufficiently addressed. However, the reviewers did note that there are still some areas of the manuscript where the grammar could be improved. For example, in section 2, the word "the" is missing in a few places before the term "mother wavelet." Also, the caption in figure 1 should be revised so that the abbreviations are in parenthesis and not quotes (see examples in captions to figures 2 and 3). Please give the manuscript a final editorial view before acceptance.*

**Reply:** Thank you very much for giving the helpful comment. We added the word "the" at some proper places in Section 2.1, and updated the caption of Figure 1 in the revised manuscript. Besides, we have carefully checked and corrected many grammatical and wording mistakes throughout the revised manuscript, mainly for making the revised manuscript more understandable and readable.

Thank you very much!

Best Regards!
Yan-Fang Sang